# 3DGS$^2$-TR: Scalable Second-Order Trust-Region Method for 3D Gaussian Splatting

**Roger Hsiao** [1]   **Yuchen Fang** [1]   **Xiangru Huang** [1]   **Ruilong Li** [2]   **Hesam Rabeti** [2]   **Zan Gojcic** [2]   **Javad Lavaei** [1]
**James Demmel** [1]   **Sophia Shao** [1]

## Abstract

We propose 3DGS$^2$-TR, a second-order optimizer for accelerating the scene training problem in 3D Gaussian Splatting (3DGS). Unlike existing second-order approaches that rely on explicit or dense curvature representations, such as 3DGS-LM (Höllein et al., 2025) or 3DGS$^2$ (Lan et al., 2025), our method approximates curvature using only the diagonal of the Hessian matrix, estimated efficiently via Hutchinson's method. Our approach is fully matrix-free and has the same complexity as ADAM (Kingma, 2014), $O(n)$ in both computation and memory costs. To ensure stable optimization in the presence of strong non-linearity in the 3DGS rasterization process, we introduce a parameter-wise trust-region technique based on the squared Hellinger distance, regularizing updates to Gaussian parameters. Under identical parameter initialization and without densification, 3DGS$^2$-TR is able to achieve better reconstruction quality on standard datasets, using 50% fewer training iterations compared to ADAM, while incurring less than 1GB of peak GPU memory overhead (17% more than ADAM and 85% less than 3DGS-LM), enabling scalability to very large scenes and potentially to distributed training settings.

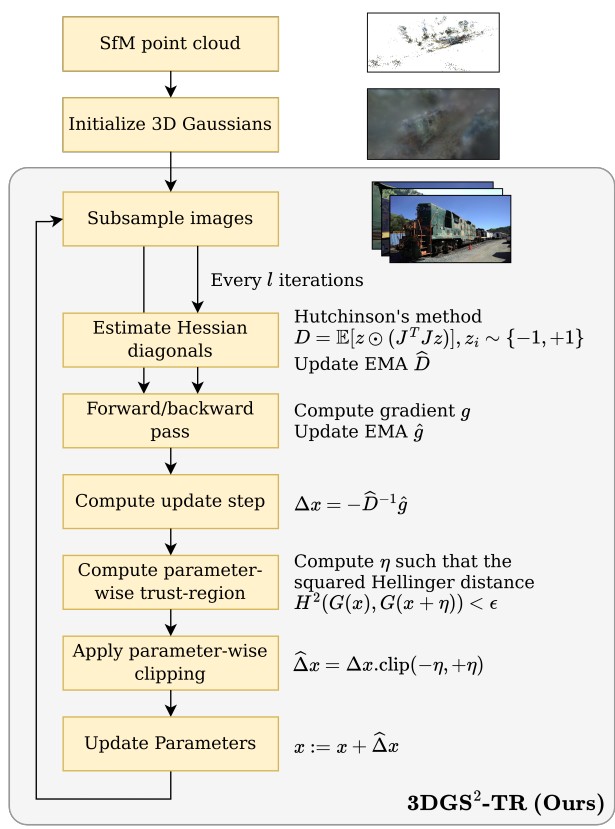

*Figure 1.* Overview of our proposed method.

## 1. Introduction

Recent advances in radiance field representations have revolutionized 3D content creation, enabling high-fidelity, photorealistic scene reconstruction from sparse input views. Neural Radiance Fields (NeRF) (Mildenhall et al., 2021) pioneered coordinate-based neural scene representations, achieving remarkable rendering quality but suffering from slow training and inference times. To address these limitations, 3D Gaussian Splatting (3DGS) (Kerbl et al., 2023) proposed an explicit and efficient scene representation that models geometry and appearance using anisotropic 3D Gaussians optimized through differentiable rasterization. This formulation preserves view-dependent effects while enabling real-time rendering, rapidly establishing 3DGS as the new standard for real-time novel view synthesis—the task of rendering photorealistic images from previously unseen camera viewpoints given a limited set of input images. This capability is central to applications in virtual reality, augmented reality, robotics, and 3D content creation.

Building on the original 3DGS framework's demonstra-

---

[1]University of California, Berkeley [2]NVIDIA. Correspondence to: Roger Hsiao <roger_hsiao@berkeley.edu>.

*Proceedings of the $43^{rd}$ International Conference on Machine Learning*, Seoul, South Korea. PMLR 306, 2026. Copyright 2026 by the author(s).

tion that anisotropic Gaussians coupled with GPU-friendly rasterization can achieve both high-quality synthesis and real-time performance (Kerbl et al., 2023), a number of subsequent works have proposed strategies to further improve rasterization efficiency. Recent efforts focus on accelerating forward rasterization and inference through optimized GPU kernels, memory access patterns, and tile-based rendering strategies, enabling efficient large-scale deployment and real-time visualization (Durvasula et al., 2023; Feng et al., 2025; Mallick et al., 2024; Ye et al., 2025; Papantonakis et al., 2024; Zhao et al., 2024).

Despite these advances in rasterization and scene representation, scene training in 3DGS remains a bottleneck in the pipeline. Although significantly faster than NeRF, 3DGS still requires training from scratch for each new scene, taking 20-40 minutes on commercial GPUs for commonly used datasets (Kerbl et al., 2023). This limits the applicability of 3DGS in large-scale or real-time exploration scenarios.

Several lines of research address this training bottleneck through orthogonal approaches. One direction targets the backward pass, which accumulates gradients into Gaussian splats with poor GPU utilization due to resource contention when many pixels share the same Gaussian primitives. Recent work has identified this as a dominant bottleneck and proposed alternative accumulation schemes or kernel redesigns to mitigate atomic contention (Durvasula et al., 2023; Ye et al., 2025). Another approach reduces training time through more compact scene representations. Related efforts improve the densification process or introduce pruning strategies to reduce the total number of Gaussian primitives while preserving reconstruction quality, thereby indirectly accelerating training speed (Kheradmand et al., 2024; Fang & Wang, 2024; Rota Bulò et al., 2024; Ali et al., 2024; Zhang et al., 2024b; Hanson et al., 2025b).

Our work aligns most closely with a third direction: improving the optimizer itself. The standard optimizer for 3DGS is ADAM (Kingma, 2014), a first-order gradient-descent method widely used in deep learning. While first-order optimizers are easy to implement and scale well, they suffer from slow convergence in highly non-convex and ill-conditioned parameter spaces (Sutton, 1986; Dauphin et al., 2014; Bottou et al., 2018). The optimization landscape of 3DGS is particularly challenging due to the strong coupling between geometry parameters (position, rotation, scale) and appearance parameters (opacity, color), leading to inefficient convergence and excessive training iterations. Additionally, first-order methods require meticulous tuning of learning rates for each parameter (Sutton, 1986; Schaul et al., 2013; Dauphin et al., 2014; Bottou et al., 2018).

3DGS occupies a unique position in machine learning: the model quality scales with the number of parameters, yet each parameter remains highly interpretable as a compo-

nent of an unnormalized 3D Gaussian distribution. This has motivated prior work to explore second-order optimization algorithms—such as Newton's method, Gauss-Newton, or Levenberg-Marquardt—to achieve superlinear convergence (Höllein et al., 2025; Lan et al., 2025; Pehlivan et al., 2025). However, the 3DGS rasterization function poses fundamental challenges for second-order methods. First, it is highly nonlinear due to the sequential rendering of Gaussian splats at each pixel, where the transmittance seen by each splat depends on the opacities of all preceding splats in depth order. Second, the loss function is only piecewise continuous since depth-based sorting of Gaussian splats causes render order to change discontinuously with position parameters. These discontinuities become more frequent in regions with dense clusters of splats, which commonly occur towards the end of training. Moreover, existing implementations of second-order methods incur high memory overhead from storing curvature information and suffer from expensive per-iteration costs due to matrix operations, offsetting any convergence speedup and making them unsuitable for production use with large scenes.

In practice, second-order optimizers for 3DGS have yet to outperform first-order methods in either reconstruction quality or convergence speed, leading to limited adoption. However, recent work in deep learning, such as Sophia (Liu et al., 2023), introduces a lightweight, second-order approach that utilizes a diagonal Hessian estimate to account for loss surface curvature. By adapting step sizes to sharp or flat regions with minimal computational overhead, Sophia achieves faster convergence and superior stability in large-scale non-convex tasks such as large language model training. Nevertheless, the performance of Sophia and its variants remains largely unexplored in computer vision applications.

Based on these observations, we propose the following three principles to accelerate scene training in 3DGS.

1. *Cheap per-iteration computation.* Since 3DGS rasterization is highly nonlinear with frequent discontinuities, we favor many small, inexpensive steps over a few large, expensive ones that would likely violate local linear or quadratic approximations. Given that the backward pass is the primary training bottleneck, our strategy introduces modest computational overhead in pre- or post-processing the update step to accelerate convergence and reduce the total number of backward passes required.

2. *Parameter-linear memory scaling.* Persistent storage across iterations should scale with the number of Gaussian parameters, not with the number of pixels, which often exceed the parameter count by orders of magnitude for large scenes with high-resolution training images.

3. *Trust region constraints.* Update step sizes should be bounded by a trust region that limits the impact of nonlinear interactions between Gaussian splats. Specifically, we bound the change in transmittance experienced by each splat before and after the update, ensuring steps remain within the region of validity for local approximation.

Guided by these principles, we present 3DGS$^2$-TR, a second-order optimizer for accelerated 3DGS scene training. We summarize our main contributions as follows:

1. We apply a second-order update rule that uses Hessian diagonals to 3DGS training, which has the same $O(n)$ complexity as ADAM in both computation and memory.

2. We propose an effective trust region for Gaussian parameter updates that bounds the squared Hellinger distance of each Gaussian splat before and after optimization steps, providing a principled constraint on geometric changes and requiring only a single tunable hyperparameter.

## 2. Related Work

### 2.1. Novel View Synthesis

Novel view synthesis aims to generate photorealistic images from arbitrary viewpoints given a set of input images. Neural Radiance Fields (NeRF) (Mildenhall et al., 2021) pioneered implicit scene representations using multilayer perceptrons to encode volumetric density and view-dependent appearance, achieving photorealistic results through volumetric ray marching but suffering from slow training and inference. In contrast, 3D Gaussian Splatting (3DGS) (Kerbl et al., 2023) models scenes explicitly as collections of anisotropic 3D Gaussians, enabling real-time rendering through efficient differentiable rasterization while maintaining high visual fidelity. This has established 3DGS as the state-of-the-art for real-time novel view synthesis, inspiring extensive research on improving rendering quality (Lu et al., 2024; Yu et al., 2024), scaling to larger environments (Kerbl et al., 2024; Song et al., 2024), and enhancing geometric accuracy.

### 2.2. Accelerating 3DGS Training and Rendering

While 3DGS achieves real-time rendering, training remains a bottleneck for large-scale scenes. Recent work addresses this through complementary strategies.

**Compact representations.** Several methods aim to reduce the number of Gaussian primitives while preserving quality. EAGLES (Girish et al., 2024) uses quantized embeddings and coarse-to-fine training. LightGaussian (Fan et al., 2024) prunes low-contribution Gaussians and compresses spherical harmonics. C3DGS (Lee et al., 2024) learns binary masks to remove redundant primitives, while Speedy-Splat (Hanson et al., 2025a) introduces tight bounding boxes and dual pruning strategies.

**Rasterization optimizations.** AdR-Gaussian (Wang et al., 2024) culls low-opacity Gaussian-tile pairs and balances workloads. FlashGS (Feng et al., 2025) and gsplat (Ye et al., 2025) optimize CUDA kernels and memory access patterns. Additional improvements include modified densification heuristics (Fang & Wang, 2024; Kheradmand et al., 2024) that reduce Gaussian proliferation.

### 2.3. Second-Order Optimizers

Recent research in large-scale machine learning explores replacing ADAM (Kingma, 2014) with second-order methods to accelerate convergence. Classical second-order methods—Newton's method, Gauss-Newton, and Levenberg-Marquardt—achieve superlinear convergence via Hessian curvature information but face $O(n^2)$ memory and $O(n^3)$ computational costs. Lightweight alternatives such as Ada-Hessian (Yao et al., 2021) and Sophia (Liu et al., 2023), instead use diagonal Hessian approximations for efficient large-scale optimization, avoiding the storage and computation overhead while gaining improvement in performance.

Applications of second-order optimizers have been explored in 3DGS as well. For example, 3DGS-LM (Höllein et al., 2025) integrates the Levenberg-Marquardt method, while 3DGS2 (Lan et al., 2025) partitions parameters and solves small Newton systems in sequence. Recently, (Pehlivan et al., 2025) adopted a matrix-free design, applying the preconditioned conjugate gradient method to solve the system, and introduced a pixel sampling strategy based on residuals.

However, some fundamental challenges have not been adequately addressed: 3DGS rasterization is highly nonlinear and only piecewise continuous due to depth-based sorting, violating smoothness assumptions of classical second-order methods. Moreover, the high parameter and pixel counts make it intractable to materialize full matrices, necessitating careful camera perspective grouping and image subsampling (Höllein et al., 2025; Lan et al., 2025). Consequently, existing approaches only work well on small scenes or when initialized near a sufficiently good local minimum. In contrast, our work adapts Sophia's lightweight framework to 3DGS with modifications tailored to Gaussian-based representations while maintaining efficiency and broad applicability.

## 3. Background and Notations

### 3.1. Review of Gaussian Splatting

3D Gaussian Splatting (3DGS) (Kerbl et al., 2023) represents a scene as a collection of 3D Gaussians splats, each parametrized by its position $\mu$, scale $S$, rotation $R$, opacity $\alpha$, and color $\mathcal{C}$. We can write each Gaussian as an unnormalized Gaussian distribution

$$G(z) = \alpha \mathcal{C} \exp(-\frac{1}{2}(z - \mu)^T \Sigma^{-1}(z - \mu)) \qquad (1)$$

$$= \alpha \mathcal{C}(2\pi)^{\frac{3}{2}} \det(\Sigma)^{\frac{1}{2}} \mathcal{N}(z; \mu, \Sigma) \qquad (2)$$

The covariance is split into separate rotation and scaling parameters as $\Sigma = R^T S^T S R$. $\mathcal{C}$ is the view-dependent color modeled using spherical harmonic coefficients of order 3. To render an image of size $H \times W$ from a given viewpoint, all Gaussians are first projected into 2D Gaussian splats via a tile-based differentiable rasterizer. The projected splats are then $\alpha$-blended along each camera ray to obtain the color $C_\omega$ at pixel $\omega$:

$$C_\omega = \sum_{i \in \mathcal{K}} \mathcal{C}_i \bar{\alpha}_i T_i, \text{ with } T_i = \prod_{j=1}^{i-1}(1 - \bar{\alpha}_j),$$

where $\mathcal{K}$ denotes the set of Gaussian kernels of size $|\mathcal{K}|$, $\mathcal{C}_i$ is the color of the $i$-th splat along the ray, $\bar{\alpha}_i$ is the 2D Gaussian's evaluated opacity, and $T_i$ represents transmittance. The complete Gaussian parameter vector $x$ concatenates the parameters of all $|\mathcal{K}|$ kernels by groups, (e.g. $x = (x_{position}, x_{scaling}, x_{rotation}, x_{opacity}, x_{color})$. To fit the Gaussian parameter $x$, 3DGS minimizes the discrepancy between the rendered and ground-truth images. At each pixel $\omega$, the loss function is written as

$$\mathcal{L}_\omega = (1 - \lambda)\mathcal{L}_{\omega, L_1} + \lambda \mathcal{L}_{\omega, D-SSIM}. \qquad (3)$$

The L1 and SSIM losses can each be further split into three components, one for each RGB channel. The total loss is the mean of the loss over all pixels and all channels.

### 3.2. Optimization Problem Formulation

Let $x \in \mathbb{R}^n$ be the parameter vector by concatenating the parameters of all Gaussian kernels. Let $M$ be the number of training images. Each training image emits $6 \times H \times W$ loss components with a total of $m$ loss components over all images. We use $f_i(x)$ to denote the vector loss function for image $i$, where each entry is the square root of a loss component, so that the 3DGS scene training problem can be formulated as a nonlinear least-squares problem

$$\min_x f(x) := \frac{1}{2m} \left\| \begin{bmatrix} f_1(x)^T & \cdots & f_M(x)^T \end{bmatrix}^T \right\|_2^2. \qquad (4)$$

The full Jacobian matrix is denoted as $J(x) = \begin{bmatrix} J_1(x)^T & \cdots & J_M(x)^T \end{bmatrix}^T$ and the pseudo-Hessian matrix is $H(x) = J(x)^T J(x) = \sum_{i=1}^M J_i(x)^T J_i(x)$.

---

**Algorithm 1** $3DGS^2$-TR

**Require:** Initial parameters $x_1 \in \mathbb{R}^n$, Hessian diagonals update interval $l \in \mathbb{N}$, EMA decay rates $\theta_1, \theta_2 \in (0, 1)$, trust-region parameter $\epsilon > 0$, Hutchinson sample size $\nu \in \mathbb{N}$.

1: Set $\widehat{g}_0 = 0, \widehat{D}_{1-l} = 0$.
2: **for** $t = 1$ to $T$ **do**
3:   Sample a mini-batch $\mathcal{S}_1$ of images and compute the stochastic gradient $g_t$.
4:   Update EMA $\widehat{g}_k = \theta_1 \widehat{g}_{k-1} + (1 - \theta_1)g_k$.
5:   **if** $t \mod l = 1$ **then**
6:     Sample a mini-batch $\mathcal{S}_2$ of images and compute $D_t = \text{Hutch}(x_t, \mathcal{S}_2, \nu)$.
7:     Update EMA $\widehat{D}_t = \theta_2 \widehat{D}_{t-l} + (1 - \theta_2)D_t$.
8:   **else**
9:     $\widehat{D}_t = \widehat{D}_{t-1}$.
10:  **end if**
11:  Compute the update step $\Delta x_t = -\widehat{D}_t^{-1}\widehat{g}_t$.
12:  Compute the parameter-wise trust-region radius

$$\eta = \text{SHD}(x_t, \epsilon).$$

13:  Apply parameter-wise clipping

$$\widehat{\Delta} x_t = \Delta x_t.\text{clip}(-\eta, +\eta).$$

14:  Update the parameters $x_{t+1} = x_t + \widehat{\Delta} x_t$.
15: **end for**

---

## 4. Methodology

$3DGS^2$-TR is a scalable second-order optimization framework for solving the 3DGS training objective (4). Our method adopts a stochastic Gauss–Newton formulation and incorporates curvature information while remaining memory- and compute-efficient. An overview of the proposed pipeline is illustrated in Figure 1, whereas the full algorithm is summarized in Algorithm 1.

### 4.1. Algorithm Design

We initialize the 3D Gaussian primitives using a sparse point cloud obtained from structure-from-motion (SfM). Given the parameters $x_t$ at iteration $t$, we first subsample a mini-batch of images $\mathcal{S}_1$ to estimate the stochastic gradient

$$g_t = \frac{1}{m} \frac{M}{|\mathcal{S}_1|} \sum_{i \in \mathcal{S}_1} J_i(x_t)^T f_i(x_t),$$

where $f_i(x_t)$ denotes the vector loss and $J_i(x_t)$ is its Jacobian of the $i$-th image with respect to the Gaussian parameters $x_t$. To reduce gradient noise and improve stability, we maintain an exponential moving average (EMA) of the

gradients with decay rate $\theta_1 \in (0, 1)$,

$$\widehat{g}_t = \theta_1 \widehat{g}_{t-1} + (1 - \theta_1)g_t.$$

To incorporate second-order information while maintaining $O(n)$ memory and computation costs, we follow the spirit of Sophia (Liu et al., 2023) and estimate the diagonal of the Gauss–Newton matrix $J^T J$ using Hutchinson's method. Specifically, every $l$ iterations, we subsample another mini-batch of images $\mathcal{S}_2$ and compute

$$D_t = \text{Hutch}(x_t, \mathcal{S}_2, \nu)$$
$$= \frac{1}{m} \frac{M}{|\mathcal{S}_2|} \frac{1}{\nu} \sum_{i=1}^{\nu} z^{(i)} \odot \left( \sum_{j \in \mathcal{S}_2} J_j(x_t)^T J_j(x_t) z^{(i)} \right),$$

where $\{z^{(i)}\}_{i=1}^m$ are independent Rademacher random vectors and $\odot$ denotes element-wise multiplication. The scaling factors $M|\mathcal{S}_1|^{-1}, M|\mathcal{S}_2|^{-1}$ are required to maintain an unbiased estimate of the gradient and the Hessian diagonal after sampling. Such image subsampling and diagonal approximation significantly reduce computational and memory overhead, since the Hessian-vector product $J_j(x_t)^T J_j(x_t)z^{(i)}$ corresponds to exactly one forward and backward pass on a training image. To further stabilize curvature estimates, we maintain an EMA of $D_t$ with decay rate $\theta_2$ and reuse the previous estimate in intermediate iterations, i.e.,

$$\widehat{D}_t = \begin{cases} \theta_2 \widehat{D}_{t-1} + (1 - \theta_2)D_t & \text{if } t \mod l = 1, \\ \widehat{D}_{t-1} & \text{otherwise.} \end{cases}$$

Given $\widehat{D}_t$ and $\widehat{g}_t$, we compute the update step

$$\Delta x_t = -\widehat{D}_t^{-1} \widehat{g}_t.$$

To improve robustness under strong nonlinearity and discontinuities in the 3DGS rasterization process, we apply a parameter-wise trust-region constraint. Specifically, we compute a trust-region radius for each parameter

$$\eta = \text{SHD}(x_t, \epsilon),$$

based on the squared Hellinger distance, which bounds the change of each Gaussian primitive induced by the update. Details of this construction are provided in Section 4.2.

The final update is obtained by parameter-wise clipping,

$$\widehat{\Delta} x_t = \Delta x_t.\text{clip}(-\eta, +\eta),$$

followed by the parameter update

$$x_{t+1} = x_t + \widehat{\Delta} x_t. \tag{5}$$

The minibatch sizes are selected based on the tradeoff between algorithmic performance and computational cost. In our experiments, we set $|\mathcal{S}_1| = |\mathcal{S}_2| = \nu = 1$ and $l = 10$, which corresponds to an approximately 10% overhead compared to ADAM.

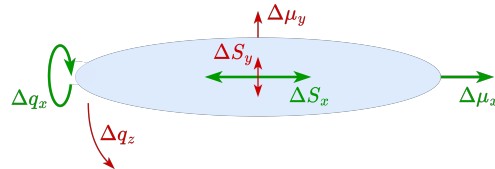

Figure 2. Example scene with a single elongated Gaussian splat. The $x$-axis and $y$-axis lie on the page; the $z$-axis comes out of the page. The green arrows mark the directions in which the Gaussian has more freedom; while the red arrows denote otherwise.

## 4.2. Trust-Region Based on Squared Hellinger Distance

As in Sophia (Liu et al., 2023), individual update steps can occasionally become excessively large due to the high variance of Hutchinson's estimator. This issue is further exacerbated in 3DGS by the highly nonlinear and piecewise-continuous nature of the rasterization process, particularly in later optimization stages where dense clusters of Gaussian primitives frequently emerge.

In ADAM (Kingma, 2014), this instability is typically addressed through careful learning-rate tuning and scheduled decay of the position updates. On the other hand, 3DGS-LM (Höllein et al., 2025) mitigates nonlinearity by introducing an adaptive regularization term that effectively interpolates between Gauss–Newton and gradient descent updates. However, both approaches operate at a global or group level and do not exploit the strong interpretability of individual Gaussian parameters.

We instead propose a *parameter-wise trust-region* strategy that leverages the explicit geometric and photometric meaning of each parameter in 3DGS. Since interactions between Gaussian splats primarily occur through their projected opacity $\bar{\alpha}$ on the image plane, a natural trust-region design should directly limit changes to this quantity across viewpoints.

To build intuition, consider a scene containing a single anisotropic Gaussian (see Figure 2). Intuitively, the Gaussian should translate or expand more conservatively along its short axis to avoid abrupt changes in rendered pixels. Additionally, rotations about the long axis should be less restrictive than those about the short axis. Finally, more transparent Gaussians should be allowed to evolve more rapidly than highly opaque ones. All of these constraints can be formalized by constructing parameter-wise trust regions that bound the discrepancy between the Gaussian splat before and after an update.

**Squared Hellinger distance.** We quantify the magnitude of an update to a Gaussian $G$ using the squared Hellinger distance

$$H^2(G, G') = \frac{1}{2} \int \left( \sqrt{G(z)} - \sqrt{G'(z)} \right)^2 dz. \tag{6}$$

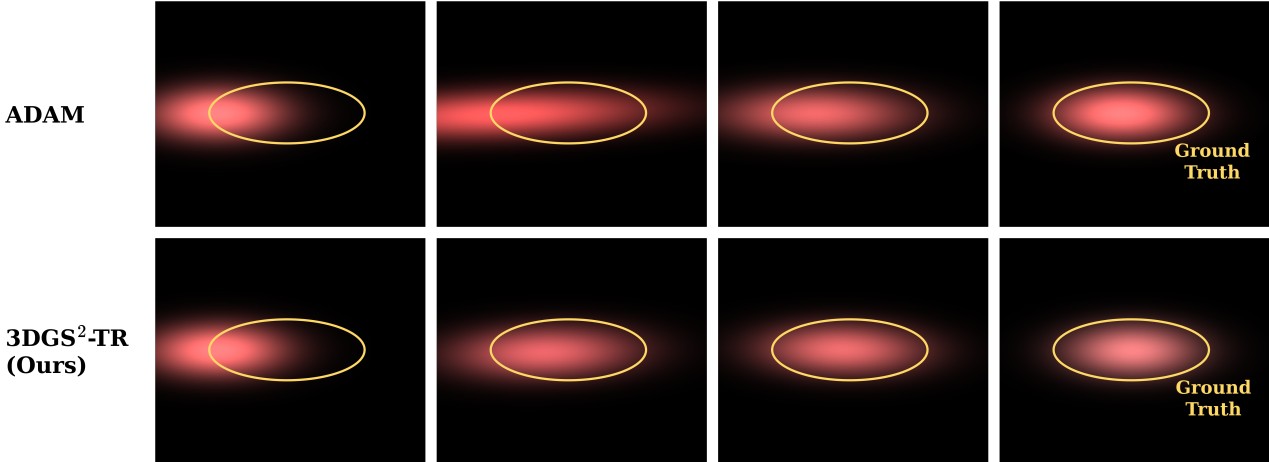

*Figure 3.* A single Gaussian fitting example where the Gaussian splat is initialized with a small perturbation. Images from left to right show the progression of optimization to fit the perturbed splat to the ground truth splat (denoted with the orange circle). Both ADAM and 3DGS$^2$-TR eventually recover the Gaussian parameters close to the ground truth, but 3DGS$^2$-TR deforms the Gaussian much less due to the trust-region bound, which increases the stability of training.

For unnormalized Gaussian primitives $G = Z \cdot \mathcal{N}(z; \mu, \Sigma)$ and $G' = Z' \cdot \mathcal{N}(z; \mu', \Sigma')$ with probability mass $Z, Z'$, $H^2(G, G')$ emits a closed-form solution

$$H^2(G, G') = \frac{1}{2}(Z + Z')$$
$$- (Z \cdot Z')^{\frac{1}{2}} \cdot \Sigma_1 \cdot \exp\left(-\frac{\Delta\mu^T \Sigma_2^{-1} \Delta\mu}{8}\right),$$

where $\Delta\mu = \mu - \mu'$, $\Sigma_1 = \frac{\det(\Sigma)^{\frac{1}{4}} \det(\Sigma')^{\frac{1}{4}}}{\det\left(\frac{\Sigma + \Sigma'}{2}\right)^{\frac{1}{2}}}$, and $\Sigma_2 = \frac{\Sigma + \Sigma'}{2}$.

In the 3DGS setting, the probability mass satisfies $Z = \alpha \mathcal{C} \det(\Sigma)^{1/2}$ and $Z' = \alpha' \mathcal{C}' \det(\Sigma')^{1/2}$. While alternative divergence measures (e.g., KL divergence) could also be employed, we find that the squared Hellinger distance yields trust-region bounds that are both simple to compute and intuitively interpretable, being the integral of the coordinate-wise difference between two Gaussian splats.

**Scale normalization.** Because the apparent mass of a rendered Gaussian can be arbitrarily scaled by moving along the viewing direction, we normalize the squared Hellinger distance by the determinant of the scale matrix. Specifically, we rescale $H^2(G, G')$ by $\det(\Sigma)^{-1/2} = \det(S)^{-1}$, recalling that $\Sigma = R^T S^T S R$ with orthogonal rotation matrix $R$. This normalization can be interpreted as comparing Gaussians at a fixed effective distance to the camera. For all parameters except color, we treat the opacity $\alpha$ as the total mass of the Gaussian, as only the opacity component affects future Gaussians to be rendered.

**Parameter-wise trust-region radius.** We now derive a parameter-wise trust-region radius $\eta = \mathrm{SHD}(x_t, \epsilon)$ such that, when a single parameter is perturbed while all others remain fixed, the normalized squared Hellinger distance between the Gaussian before the update $G(x_t)$ and after the update $G(x_t + \eta)$ is bounded by $\epsilon$:

$$\eta = \mathrm{SHD}(x_t, \epsilon) := \arg\max_{|\eta|} \left\{ |\eta| : \frac{H^2(G, G')}{\det(S)} < \epsilon \right\}.$$

Below, we summarize the resulting trust-region bounds for each parameter type; detailed derivations are deferred to Appendix B.

**Bound on the mean** $\mu = (\mu_x, \mu_y, \mu_z)$. For perturbations $\Delta\mu = (\Delta\mu_x, \Delta\mu_y, \Delta\mu_z)$, with $c \in \{x, y, z\}$, we require

$$|\Delta\mu_c| < \sqrt{-8 \, \Sigma_{cc} \ln\left(1 - \frac{\epsilon}{\alpha}\right)}.$$

**Bound on the scale matrix** $S$. Let

$$S = \begin{bmatrix} S_x & & \\ & S_y & \\ & & S_z \end{bmatrix}, \qquad S' = \begin{bmatrix} S'_x & & \\ & S'_y & \\ & & S'_z \end{bmatrix},$$

and define $\Delta S = S' - S$ with blockwise differences $\Delta S_x, \Delta S_y, \Delta S_z$, then with $c \in \{x, y, z\}$, we require

$$|\Delta S_c| < \sqrt{\frac{4 S_c^2 \, \epsilon}{\alpha}}.$$

**Bound on the opacity** $\alpha$. Let $\alpha' = \alpha + \Delta\alpha$. To control the mass discrepancy, we require

$$|\Delta\alpha| < \sqrt{8\alpha \, \epsilon}.$$

**Bound on the color** $\mathcal{C}$. Let $\mathcal{C} = \mathcal{C}_r + \mathcal{C}_g + \mathcal{C}_b$, $\mathcal{C}' = \mathcal{C}'_r + \mathcal{C}'_g + \mathcal{C}'_b$ and $\Delta\mathcal{C}_c = \mathcal{C}'_c - \mathcal{C}_c$ for $c \in \{r, g, b\}$. We impose

$$|\Delta\mathcal{C}_c| < \sqrt{\frac{8\mathcal{C}_c\,\epsilon}{\alpha}}.$$

**Bound on the rotation** $R$. The rotation matrix $R$ is parameterized by a quaternion $(q_x, q_y, q_z, q_w)$ with squared norm $q^2 = q_x^2 + q_y^2 + q_z^2 + q_w^2$. Let $\beta_c > 0$ denote a geometry-dependent constant that upper bounds the sensitivity of the induced Gaussian displacement with respect to perturbations in the $c$-th quaternion component (see Appendix B). To ensure the rotation update remains within the trust region, for $c \in \{x, y, z, w\}$ it suffices to require

$$|\Delta q_c| < \sqrt{-\frac{8}{\beta_c}\ln\left(1 - \frac{\epsilon}{\alpha}\right)}.$$

We illustrate the effects of the proposed Hellinger-distance-based trust region using a single Gaussian fitting example in Figure 3.

# 5. Evaluations

We now present the evaluation results of our method. The forward-mode automatic differentiation of the 3DGS rasterization kernel uses an in-house CUDA implementation based on duo numbers, which can be found at https://github.com/rogerhh/diff-gaussian-rasterization-jvp. The parameter-wise trust-region radius is implemented as a custom CUDA kernel. The training pipeline and the Sophia optimizer is implemented in PyTorch following the original 3DGS implementation.

Our experiments are run on A100-SXM4 GPUs with 6921 CUDA cores and 80 GB VRAM. We evaluate our method on the same datasets as 3DGS, namely, all scenes from MiP-NeRF360, two scenes from Tanks & Temples, and two scenes from Deep Blending (Barron et al., 2022; Knapitsch et al., 2017; Hedman et al., 2018). For each dataset, we initialize the 3D Gaussian splats with the standard SfM point cloud. As our diagonal estimator currently does not handle inserting new diagonal entries, all of our experiments are performed *without densification*.

In all experiments, we set the Hessian diagonal update interval to $l = 10$ to balance computational cost and curvature accuracy, following the default configuration of Sophia (Liu et al., 2023), and choose the exponential moving average (EMA) parameters $\theta_1 = 0.9$ and $\theta_2 = 0.999$. The trust-region radius $\epsilon$ follows a exponential decay schedule from $10^{-6}$ to $10^{-8}$ over the course of training.

For our baselines, we compare our method against ADAM, the SOTA first-order method, and 3DGS-LM, the only

second-order optimizer for 3DGS with open-source implementation.

We also include an ablation study which applies our Hellinger-distance-based trust region to the ADAM update step (ADAM-TR). On the other hand, Sophia optimizer alone without trust region tends to produce NaN values due to instability of the stochastic diagonal estimator for some splats; therefore, our method is always reported with trust region enabled.

Since 3DGS-LM has a high computation overhead, we only run it for 150 iterations in total, and report the results at 35, 75, and 150 iterations. We run all other methods for 30k iterations and report results at 7k, 15k, and 30k iterations. We use the same train/test split and report the same metrics (SSIM, PSNR, LPIPS) on the test images as proposed by 3DGS (Kerbl et al., 2023).

The main quantitative results are presented in Table 1. (Results for individual scenes can be found in Appendix A.) Qualitative comparisons for some scenes are shown in Figure 4. 3DGS$^2$-TR significantly outperforms other methods, reaching comparable or better reconstruction quality using 50% fewer iterations than first-order methods (ADAM and ADAM-TR). In the case of Tanks & Temples, 3DGS$^2$-TR is able to exceed the best ADAM PSNR by 0.56dB at 7k iterations and 1.19dB at 30k iterations. Moreover, 3DGS$^2$-TR requires less than 1GB of additional GPU memory for training in all scenes, which is 17% more than ADAM and 85% less than 3DGS-LM on average.

In contrast, 3DGS-LM does not produce comparable results with the same initialization, given significantly more time and resources. We also note that by applying our trust-region clipping to the ADAM updates (ADAM-TR), we are already able to achieve better quality than the vanilla ADAM optimizer.

# 6. Discussions and Limitations

Our method achieves much faster convergence per step compared to ADAM. However, due to a naive implementation of the Sophia optimizer in PyTorch, our second-order method has a higher run time per iteration, which is not a fundamental limitation of the algorithm itself. We plan to optimize the PyTorch implementation to reach comparable performance to ADAM for tensor manipulation in future work.

Compared to other proposed second-order optimizers, our method is fairly non-intrusive to the vanilla training pipeline, which makes it amenable to orthogonal improvements to 3DGS training, such as Markov Chain Monte Carlo (MCMC) densification (Kheradmand et al., 2024), co-regularization (Zhang et al., 2024a), drop-out Gaussians (Park et al., 2025), etc. However, one current limitation with

*Table 1.* The main quantitative result. Best results are **boldfaced** and second best are underlined.

| Method | SSIM ↓ | | | PSNR ↑ | | | LPIPS ↓ | | | Time (s) | Peak GPU Mem (GB) |
|---|---|---|---|---|---|---|---|---|---|---|---|
| | @7k | @15k | @30k | @7k | @15k | @30k | @7k | @15k | @30k | | |
| mipnerf | | | | | | | | | | | |
| ADAM | 0.667 | 0.679 | 0.684 | 24.45 | 24.85 | 25.08 | 0.406 | 0.400 | 0.420 | **212** | **5.60** |
| 3DGS-LM [1] | 0.493 | 0.520 | 0.549 | 20.02 | 21.02 | 21.88 | 0.598 | 0.577 | 0.548 | 2602 | 47.75 |
| ADAM-TR (Ours) | 0.674 | 0.688 | 0.693 | 24.54 | 25.02 | 25.21 | 0.394 | 0.388 | 0.411 | 235 | 5.64 |
| 3DGS$^2$-TR (Ours) [2] | **0.682** | **0.692** | **0.696** | **24.80** | **25.19** | **25.39** | **0.390** | **0.385** | **0.402** | 652 | 6.51 |
| deepblending | | | | | | | | | | | |
| ADAM | 0.845 | 0.854 | 0.857 | 26.65 | 27.18 | 27.36 | 0.372 | 0.365 | 0.387 | **180** | **5.42** |
| 3DGS-LM [1] | 0.810 | 0.833 | 0.845 | 23.72 | 24.89 | 25.62 | 0.437 | 0.405 | 0.385 | 2235 | 33.33 |
| ADAM-TR (Ours) | 0.855 | 0.866 | **0.869** | 26.56 | 27.34 | 27.58 | 0.357 | 0.351 | 0.375 | 195 | 5.51 |
| 3DGS$^2$-TR (Ours) [2] | **0.859** | **0.867** | 0.869 | **26.91** | **27.54** | **27.74** | **0.354** | **0.349** | **0.367** | 600 | 6.26 |
| tandt | | | | | | | | | | | |
| ADAM | 0.723 | 0.755 | 0.766 | 20.69 | 21.41 | 21.66 | 0.309 | 0.298 | 0.341 | **201** | **3.17** |
| 3DGS-LM [1] | 0.661 | 0.717 | 0.749 | 20.00 | 21.09 | 21.81 | 0.429 | 0.366 | 0.331 | 2105 | 25.53 |
| ADAM-TR (Ours) | 0.774 | 0.791 | 0.798 | 21.62 | 22.16 | 22.42 | 0.273 | 0.264 | 0.295 | 227 | 3.27 |
| 3DGS$^2$-TR (Ours) [2] | **0.787** | **0.800** | **0.805** | **22.12** | **22.63** | **22.85** | **0.262** | **0.255** | **0.279** | 484 | 4.12 |

[1] 3DGS-LM results are reported @35, @75, and @150 iterations.
[2] System implementation is not fully optimized. See Section 6 for details.

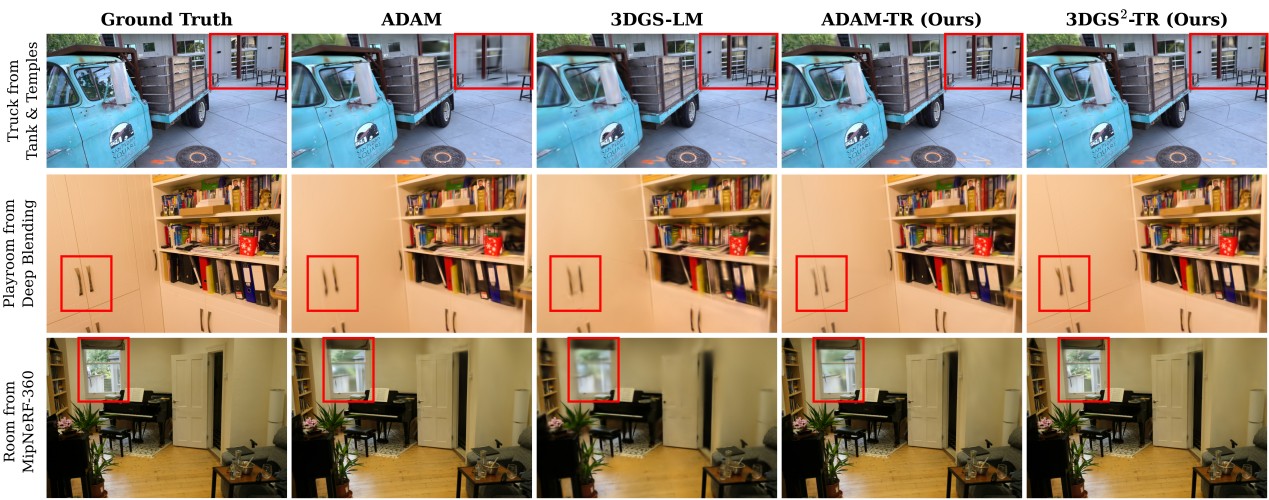

*Figure 4.* Qualitative comparison of different methods of the truck, playroom, and room scenes. The red boxes highlight regions where 3DGS$^2$-TR significantly outperforms other methods.

our work is that we do not support adding or moving around Gaussian splats. Naively inserting rows and columns to the Hessian matrix results in a biased estimation of the diagonal entries, leading to degraded performance, which we aim to resolve in future work.

## 7. Conclusion

We present 3DGS$^2$-TR, a second-order optimizer for training 3D Gaussian Splatting scenes. We approximate the curvature information of the loss function using a stochastic diagonal estimator, which eliminates the storage and computation overhead of classical second-order methods, thus maintains the same $O(n)$ complexity as ADAM in both computational cost and memory. Furthermore, we introduce a parameter-wise trust-region technique based on the squared Hellinger distance to bound the update step size. Instead of tuning learning rates for each parameter group, our method requires only one hyperparameter. We show that 3DGS$^2$-TR achieves better reconstruction quality after

50% of training iterations, while incurring only 10% of computation overhead and less than 1GB of memory overhead compared to ADAM. Our method is a drop-in replacement for the ADAM optimizer to accelerate 3DGS training in all settings.

## Acknowledgements

We thank the reviewers for their valuable feedback and comments. This research was partially funded by SLICE Lab industrial sponsors.

## Impact Statement

This paper presents work whose goal is to advance the field of Machine Learning. There are many potential societal consequences of our work, none which we feel must be specifically highlighted here.

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

# A. Quantitative Results Per Scene

*Table 2.* Results on Deep Blending. Best results are **boldfaced** and second best are underlined.

| Method | Scene | Deep Blending | | | | | | | | | Time | Peak GPU |
| | | SSIM ↓ | | | PSNR ↑ | | | LPIPS ↓ | | | (s) | Mem (GB) |
| | | @7k | @15k | @30k | @7k | @15k | @30k | @7k | @15k | @30k | | |
|---|---|---|---|---|---|---|---|---|---|---|---|---|
| ADAM | drjohnson | 0.841 | 0.855 | 0.861 | **26.60** | **27.39** | **27.64** | 0.368 | 0.360 | 0.390 | **182** | **5.82** |
| 3DGS-LM [1] | drjohnson | 0.811 | 0.837 | 0.849 | 24.78 | 25.87 | 26.37 | 0.432 | 0.393 | 0.373 | 3344 | 47.65 |
| ADAM-TR (Ours) | drjohnson | 0.852 | **0.865** | **0.869** | 26.50 | 27.30 | 27.56 | 0.355 | 0.347 | 0.378 | 200 | 5.90 |
| 3DGS$^2$-TR (Ours) [2] | drjohnson | **0.855** | 0.865 | 0.869 | 26.54 | 27.25 | 27.48 | **0.353** | **0.346** | **0.371** | 606 | 6.68 |
| ADAM | playroom | 0.849 | 0.852 | 0.854 | 26.70 | 26.97 | 27.07 | 0.376 | 0.371 | 0.385 | **177** | **5.03** |
| 3DGS-LM [1] | playroom | 0.809 | 0.829 | 0.840 | 22.66 | 23.91 | 24.87 | 0.443 | 0.417 | 0.398 | 1125 | 19.01 |
| ADAM-TR (Ours) | playroom | 0.859 | 0.867 | 0.869 | 26.63 | 27.38 | 27.61 | 0.359 | 0.354 | 0.373 | 189 | 5.12 |
| 3DGS$^2$-TR (Ours) [2] | playroom | **0.863** | **0.868** | **0.870** | **27.28** | **27.84** | **27.99** | **0.354** | **0.351** | **0.363** | 594 | 5.84 |

[1] 3DGS-LM results are reported @35, @75, and @150 iterations.
[2] System implementation is not fully optimized. See Section 6 for details.

*Table 3.* Results on Tanks & Temples. Best results are **boldfaced** and second best are underlined.

| Method | Scene | Tanks & Temples | | | | | | | | | Time | Peak GPU |
| | | SSIM ↓ | | | PSNR ↑ | | | LPIPS ↓ | | | (s) | Mem (GB) |
| | | @7k | @15k | @30k | @7k | @15k | @30k | @7k | @15k | @30k | | |
|---|---|---|---|---|---|---|---|---|---|---|---|---|
| ADAM | train | 0.697 | 0.745 | 0.758 | 19.83 | 20.91 | 21.21 | 0.318 | 0.302 | 0.362 | **209** | **3.45** |
| 3DGS-LM [1] | train | 0.644 | 0.688 | 0.720 | 19.13 | 19.92 | 20.57 | 0.425 | 0.377 | 0.347 | 2162 | 28.36 |
| ADAM-TR (Ours) | train | 0.740 | 0.763 | 0.771 | 20.21 | 20.87 | 21.22 | 0.297 | 0.287 | 0.321 | 229 | 3.54 |
| 3DGS$^2$-TR (Ours) [2] | train | **0.757** | **0.775** | **0.781** | **20.68** | **21.32** | **21.62** | **0.283** | **0.276** | **0.302** | 503 | 4.43 |
| ADAM | truck | 0.750 | 0.766 | 0.773 | 21.56 | 21.90 | 22.12 | 0.301 | 0.293 | 0.320 | **192** | **2.88** |
| 3DGS-LM [1] | truck | 0.679 | 0.747 | 0.779 | 20.86 | 22.25 | 23.05 | 0.432 | 0.355 | 0.315 | 2047 | 22.69 |
| ADAM-TR (Ours) | truck | 0.807 | 0.819 | 0.824 | 23.02 | 23.44 | 23.62 | 0.250 | 0.242 | 0.268 | 224 | 3.00 |
| 3DGS$^2$-TR (Ours) [2] | truck | **0.816** | **0.826** | **0.829** | **23.55** | **23.93** | **24.07** | **0.240** | **0.234** | **0.255** | 465 | 3.82 |

[1] 3DGS-LM results are reported @35, @75, and @150 iterations.
[2] System implementation is not fully optimized. See Section 6 for details.

*Table 4.* Results on MipNeRF-360. Best results are **boldfaced** and second best are underlined.

| Method | Scene | SSIM ↓ | | | PSNR ↑ | | | LPIPS ↓ | | | Time (s) | Peak GPU Mem (GB) |
|---|---|---|---|---|---|---|---|---|---|---|---|---|
| | | @7k | @15k | @30k | @7k | @15k | @30k | @7k | @15k | @30k | | |
| ADAM | treehill | 0.503 | 0.516 | 0.520 | 21.43 | 21.65 | 21.69 | 0.549 | 0.544 | 0.558 | **194** | 3.47 |
| 3DGS-LM [1] | treehill | 0.397 | 0.413 | 0.430 | 19.31 | 19.99 | 20.52 | 0.645 | 0.634 | 0.617 | 2648 | 42.58 |
| ADAM-TR (Ours) | treehill | 0.510 | 0.524 | 0.529 | 21.47 | 21.71 | 21.75 | 0.538 | 0.533 | 0.552 | 210 | **3.46** |
| 3DGS$^2$-TR (Ours) [2] | treehill | **0.520** | **0.530** | **0.533** | **21.73** | **21.84** | **21.84** | **0.533** | **0.529** | **0.543** | 600 | 4.26 |
| ADAM | counter | 0.852 | 0.866 | 0.872 | 26.49 | 27.15 | 27.53 | 0.279 | 0.270 | 0.297 | **241** | **7.12** |
| 3DGS-LM [1] | counter | 0.678 | 0.709 | 0.741 | 20.68 | 21.95 | 23.07 | 0.505 | 0.480 | 0.447 | 3531 | 61.56 |
| ADAM-TR (Ours) | counter | 0.859 | 0.871 | 0.876 | 26.58 | 27.25 | 27.59 | 0.272 | 0.265 | 0.290 | 270 | 7.24 |
| 3DGS$^2$-TR (Ours) [2] | counter | **0.865** | **0.874** | **0.877** | **26.91** | **27.51** | **27.77** | **0.268** | **0.263** | **0.282** | 761 | 8.18 |
| ADAM | stump | 0.508 | 0.517 | 0.520 | 22.77 | 22.83 | 22.86 | 0.542 | 0.538 | 0.550 | **182** | **2.60** |
| 3DGS-LM [1] | stump | 0.384 | 0.397 | 0.416 | 20.44 | 20.85 | 21.30 | 0.670 | 0.662 | 0.643 | 2121 | 33.61 |
| ADAM-TR (Ours) | stump | 0.512 | 0.528 | 0.532 | 22.78 | 22.93 | 22.99 | 0.529 | 0.524 | 0.546 | 187 | 2.64 |
| 3DGS$^2$-TR (Ours) [2] | stump | **0.525** | **0.534** | **0.537** | **22.98** | **23.08** | **23.13** | **0.522** | **0.518** | **0.532** | 589 | 3.39 |
| ADAM | bonsai | 0.893 | 0.904 | 0.908 | 28.37 | 29.24 | 29.69 | 0.285 | 0.279 | 0.299 | **237** | **8.49** |
| 3DGS-LM [1] | bonsai | 0.679 | 0.721 | 0.760 | 21.04 | 22.68 | 24.06 | 0.505 | 0.482 | 0.450 | 1921 | 43.60 |
| ADAM-TR (Ours) | bonsai | 0.901 | 0.911 | 0.914 | 28.75 | 29.67 | 30.03 | 0.273 | 0.268 | 0.288 | 278 | 8.63 |
| 3DGS$^2$-TR (Ours) [2] | bonsai | **0.905** | **0.913** | **0.916** | **29.11** | **29.92** | **30.31** | **0.270** | **0.265** | **0.283** | 750 | 9.59 |
| ADAM | bicycle | 0.480 | 0.498 | 0.504 | 21.58 | 21.84 | 21.95 | 0.523 | 0.516 | 0.539 | **185** | **3.70** |
| 3DGS-LM [1] | bicycle | 0.346 | 0.360 | 0.376 | 18.96 | 19.47 | 19.89 | 0.676 | 0.666 | 0.652 | 2265 | 42.61 |
| ADAM-TR (Ours) | bicycle | 0.494 | 0.516 | 0.523 | 21.70 | 22.04 | 22.11 | 0.505 | 0.496 | 0.530 | 203 | 3.76 |
| 3DGS$^2$-TR (Ours) [2] | bicycle | **0.511** | **0.526** | **0.532** | **21.92** | **22.17** | **22.24** | **0.494** | **0.488** | **0.510** | 565 | 4.53 |
| ADAM | kitchen | 0.880 | 0.891 | 0.898 | 27.95 | 28.64 | 29.11 | 0.193 | 0.185 | 0.209 | **271** | 8.41 |
| 3DGS-LM [1] | kitchen | 0.592 | 0.640 | 0.698 | 20.94 | 22.38 | 23.66 | 0.490 | 0.450 | 0.401 | 3257 | 66.58 |
| ADAM-TR (Ours) | kitchen | 0.888 | 0.897 | 0.902 | 28.03 | 28.81 | 29.20 | 0.186 | 0.179 | 0.200 | 307 | **8.35** |
| 3DGS$^2$-TR (Ours) [2] | kitchen | **0.892** | **0.899** | **0.903** | **28.51** | **29.13** | **29.47** | **0.183** | **0.177** | **0.194** | 761 | 9.51 |
| ADAM | flowers | 0.353 | 0.362 | 0.365 | 18.77 | 18.88 | 18.89 | 0.605 | 0.602 | 0.614 | **187** | 3.41 |
| 3DGS-LM [1] | flowers | 0.265 | 0.277 | 0.292 | 17.02 | 17.51 | 17.89 | 0.725 | 0.711 | 0.690 | 2654 | 48.45 |
| ADAM-TR (Ours) | flowers | 0.363 | 0.374 | 0.377 | 18.81 | 18.94 | 18.97 | 0.594 | 0.590 | 0.605 | 202 | **3.39** |
| 3DGS$^2$-TR (Ours) [2] | flowers | **0.370** | **0.377** | **0.379** | **18.93** | **19.02** | **19.04** | **0.592** | **0.589** | **0.599** | 555 | 4.18 |
| ADAM | room | 0.870 | 0.879 | 0.883 | 28.80 | 29.33 | 29.67 | 0.309 | 0.302 | 0.323 | **212** | **8.82** |
| 3DGS-LM [1] | room | 0.718 | 0.766 | 0.797 | 21.96 | 23.79 | 25.32 | 0.488 | 0.467 | 0.438 | 2696 | 49.79 |
| ADAM-TR (Ours) | room | 0.877 | 0.886 | 0.890 | 28.77 | 29.49 | 29.81 | 0.296 | 0.290 | 0.312 | 238 | 8.91 |
| 3DGS$^2$-TR (Ours) [2] | room | **0.881** | **0.888** | **0.891** | **29.00** | **29.70** | **30.18** | **0.292** | **0.286** | **0.304** | 691 | 9.72 |
| ADAM | garden | 0.660 | 0.678 | 0.687 | 23.87 | 24.11 | 24.29 | 0.370 | 0.363 | 0.387 | **202** | **4.36** |
| 3DGS-LM [1] | garden | 0.373 | 0.394 | 0.428 | 19.88 | 20.53 | 21.23 | 0.674 | 0.644 | 0.595 | 2327 | 40.97 |
| ADAM-TR (Ours) | garden | 0.664 | 0.685 | 0.692 | 23.98 | 24.32 | 24.45 | 0.356 | 0.349 | 0.379 | 217 | 4.42 |
| 3DGS$^2$-TR (Ours) [2] | garden | **0.672** | **0.688** | **0.694** | **24.12** | **24.37** | **24.50** | **0.353** | **0.346** | **0.369** | 591 | 5.26 |

[1] 3DGS-LM results are reported @35, @75, and @150 iterations.
[2] System implementation is not fully optimized. See Section 6 for details.

## B. Parameter-wise trust-region bounds based on squared Hellinger distance

We derive parameter-wise trust-region bounds such that when one parameter is updated while all other parameters fixed, the normalized distance between the Gaussian primitive before and after the update guarantees

$$\frac{H^2(G, G')}{\det(S)} < \epsilon.$$

**Bound on the mean $\mu$.** Let $\mu = (\mu_x, \mu_y, \mu_z), \mu' = (\mu'_x, \mu'_y, \mu'_z), \Delta\mu = \mu' - \mu$. We first let $\Delta\mu_x \neq 0, \Delta\mu_y = 0, \Delta\mu_z = 0$, then

$$H^2(G, G') = C \left(1 - \exp\left(-\frac{\Delta\mu^T \Sigma^{-1} \Delta\mu}{8}\right)\right) = C \left(1 - \exp\left(-\frac{\Delta\mu_x^2}{8\Sigma_{xx}}\right)\right).$$

Setting $H^2(G, G')/\det(S) < \epsilon$ gives $\alpha \left(1 - \exp\left(-\frac{\Delta\mu_x^2}{8\Sigma_{xx}}\right)\right) < \epsilon$. After rearranging the terms, we have

$$|\Delta\mu_x| < \sqrt{-8\,\Sigma_{xx}\,\ln\left(1 - \frac{\epsilon}{\alpha}\right)}.$$

Using similar derivation, we have

$$|\Delta\mu_y| < \sqrt{-8\,\Sigma_{yy}\,\ln\left(1 - \frac{\epsilon}{\alpha}\right)},$$

$$|\Delta\mu_z| < \sqrt{-8\,\Sigma_{zz}\,\ln\left(1 - \frac{\epsilon}{\alpha}\right)}.$$

**Bound on the scale matrix $S$.** Let

$$S = \begin{bmatrix} S_x & & \\ & S_y & \\ & & S_z \end{bmatrix}, \qquad S' = \begin{bmatrix} S'_x & & \\ & S'_y & \\ & & S'_z \end{bmatrix},$$

and define $\Delta S = S' - S$ with differences $\Delta S_x, \Delta S_y, \Delta S_z$.

Note that $C = \alpha\mathcal{C}\det(\Sigma)^{1/2} = \alpha\mathcal{C}\det(S), C' = \alpha\mathcal{C}\det(S')$, we first let $\Delta S_x \neq 0$ while $\Delta S_y = 0, \Delta S_z = 0$, and denote $\rho_x = \alpha S_y S_z$. Then

$$H^2(G, G') = \frac{1}{2}(\rho_x S_x + \rho_x S'_x) - \rho_x (S_x S'_x)^{1/2} \frac{(S_x)^{\frac{1}{2}}(S'_x)^{\frac{1}{2}} S_y S_z}{(\frac{S_x^2 + S_x'^2}{2})^{\frac{1}{2}} S_y S_z} = \rho_x \frac{S_x + S'_x}{2} - \rho_x \frac{S_x S'_x}{\left(\frac{S_x^2 + S_x'^2}{2}\right)^{\frac{1}{2}}}$$

Since $S'_x = S_x + \Delta S_x$, we have $\frac{d}{d\Delta S_x} H^2 = \rho_x \left(\frac{1}{2} + \frac{-2S_x^2}{(2S_x^2 + \Delta S_x)^2}\right)$, and $\frac{d^2}{d\Delta S_x^2} H^2 = \rho_x (4S_x^2 \cdot (2S_x + \Delta S_x)^{-3})$. We thus can approximate

$$H^2(G, G') \approx \frac{1}{2}\frac{\rho_x}{2S_x}\Delta S_x^2.$$

Setting $H^2(G, G')/\det S < \epsilon$ gives

$$|\Delta S_x| < \sqrt{\frac{4S_x \det(S)\epsilon}{\rho_x}} = \sqrt{\frac{4S_x^2 \epsilon}{\alpha}}.$$

We can similarly obtain

$$|\Delta S_y| < \sqrt{\frac{4S_y^2 \epsilon}{\alpha}}, \text{ and } |\Delta S_z| < \sqrt{\frac{4S_z^2 \epsilon}{\alpha}}.$$

**Bound on the opacity $\alpha$.** Let $\rho_\alpha = \det(\Sigma)^{1/2} = \det(S)$, then

$$H^2(G, G') = \rho_\alpha \left( \frac{\alpha + \alpha'}{2} - (\alpha\alpha')^{1/2} \right).$$

Let $\alpha' = \alpha + \Delta\alpha$, then

$$\frac{\mathrm{d}}{\mathrm{d}\Delta\alpha} H^2 = \rho_\alpha \left( \frac{1}{2} - \frac{1}{2}\alpha(\alpha^2 + \alpha\Delta\alpha)^{-1/2} \right)$$

and

$$\frac{\mathrm{d}^2}{\mathrm{d}\Delta\alpha^2} H^2 = \rho_\alpha \left( \frac{1}{4}\alpha^2(\alpha^2 + \alpha\Delta\alpha)^{-3/2} \right).$$

Thus, we can approximate

$$H^2(G, G') \approx \frac{1}{2} \frac{\rho_\alpha}{4\alpha} \Delta\alpha^2.$$

Setting $H^2(G, G')/\det(S) < \epsilon$ gives

$$|\Delta\alpha| < \sqrt{\frac{8\alpha \det(S)\epsilon}{\rho_\alpha}} = \sqrt{8\alpha\epsilon}.$$

**Bound on the color $\mathcal{C}$.** Let $\mathcal{C} = \mathcal{C}_r + \mathcal{C}_g + \mathcal{C}_b$ and similarly define $\mathcal{C}'$ and $\Delta\mathcal{C}_c = \mathcal{C}'_c - \mathcal{C}_c$ for $c \in \{r, g, b\}$. First, let $\Delta\mathcal{C}_r \neq 0$ and $\Delta\mathcal{C}_g = \Delta\mathcal{C}_b = 0$. Let $\rho_r = \alpha \det(\Sigma)^{1/2} = \alpha \det(S)$, then

$$H^2(G, G') = \rho_r \left( \frac{1}{2}(\mathcal{C}_r + \mathcal{C}'_r) - (\mathcal{C}_r\mathcal{C}'_r)^{1/2} \right).$$

We can approximate

$$H^2(G, G') \approx \frac{1}{2} \frac{\rho_r}{4\mathcal{C}_r} \Delta\mathcal{C}_r^2.$$

Setting $H^2(G, G')/\det(S) < \epsilon$ gives

$$|\Delta\mathcal{C}_r| < \sqrt{\frac{8\mathcal{C}_r \det(S)\epsilon}{\rho_r}} = \sqrt{\frac{8\mathcal{C}_r\epsilon}{\alpha}}.$$

Using similar derivation, we also have

$$|\Delta\mathcal{C}_g| < \sqrt{\frac{8\mathcal{C}_g\,\epsilon}{\alpha}}, \ |\Delta\mathcal{C}_b| < \sqrt{\frac{8\mathcal{C}_b\,\epsilon}{\alpha}}.$$

**Bound on the rotation $R$.** The rotation matrix $R$ is parameterized by a unnormalized quaternion $\tilde{q} = (q_x, q_y, q_z, q_w)$ with squared norm $r^2 = \|q\|^2 = q_x^2 + q_y^2 + q_z^2 + q_w^2$, yielding

$$R = \begin{bmatrix} 1 - \frac{2(q_y^2 + q_z^2)}{r^2} & \frac{2(q_x q_y - q_w q_z)}{r^2} & \frac{2(q_x q_z + q_w q_y)}{r^2} \\ \frac{2(q_x q_y + q_w q_z)}{r^2} & 1 - \frac{2(q_z^2 + q_x^2)}{r^2} & \frac{2(q_y q_z - q_w q_x)}{r^2} \\ \frac{2(q_x q_z - q_w q_y)}{r^2} & \frac{2(q_y q_z + q_w q_x)}{r^2} & 1 - \frac{2(q_x^2 + q_y^2)}{r^2} \end{bmatrix}.$$

We can similarly construct $R'$, which is parameterized by another unnormalized quaternion $q' = (q'_x, q'_y, q'_z, q'_w)$ with squared norm $r'^2 = \|q'\|^2 = q_x'^2 + q_y'^2 + q_z'^2 + q_w'^2$.

Since $\Sigma = R^T S^T S R$, $\Sigma' = R'^T S^T S R'$ and $R, R'$ are orthogonal, we have $\det(\Sigma) = \det(\Sigma') = \det(S)^2$. Let $\Delta R = R^{-1}R'$ also be an orthogonal rotation matrix. Then

$$H^2(G, G') = C \left( 1 - \frac{\det(S)}{\det(\frac{\Sigma + \Sigma'}{2})^{\frac{1}{2}}} \right) = C \left( 1 - \frac{\det(S)}{\det(\frac{S^2 + \Delta R^T S^2 \Delta R}{2})^{\frac{1}{2}}} \right).$$

Setting $H^2(G, G')/\det(S) < \epsilon$ and rearranging the terms gives

$$1 - \frac{\det(S)}{\det\left(\frac{S^2 + \Delta R^T S^2 \Delta R}{2}\right)^{\frac{1}{2}}} < \frac{\epsilon}{\alpha}.$$

To satisfy the above inequality, it is sufficient that $\det\left(\frac{S^2 + \Delta R^T S^2 \Delta R}{2}\right) < \left(\frac{\det(S)}{1 - \epsilon/\alpha}\right)^2$, which is sufficient if $\det\left(S^2\left(\frac{I + S^{-2}\Delta R^T S^2 \Delta R}{2}\right)\right) < \left(\frac{\det(S)}{1 - \epsilon/\alpha}\right)^2$. Since for two matrices $A$ and $B$, $\det(AB) = \det(A)\det(B)$, we only need $\det\left(\frac{I + S^{-2}\Delta R^T S^2 \Delta R}{2}\right) < \left(\frac{1}{1 - \epsilon/\alpha}\right)^2$, equivalently,

$$\det\left(I + \left(-I + \frac{I + S^{-2}\Delta R^T S^2 \Delta R}{2}\right)\right) < \left(\frac{1}{1 - \epsilon/\alpha}\right)^2.$$

By the relation $\det(I + X) \le \exp(\mathrm{tr}(X))$, it is sufficient to have

$$\mathrm{tr}\left(\frac{I + S^{-2}\Delta R^T S^2 \Delta R}{2} - I\right) < -2\ln\left(1 - \frac{\epsilon}{\alpha}\right).$$

By the linearity of trace, we only need to show

$$\mathrm{tr}(S^{-2}\Delta R^T S^2 \Delta R) - 3 < -4\ln\left(1 - \frac{\epsilon}{\alpha}\right). \tag{7}$$

Next, we investigate the approximation of $\mathrm{tr}(S^{-2}\Delta R^T S^2 \Delta R)$. For an unnormalized quaternion $\tilde{q} = (q_x, q_y, q_z, q_w)$ with $r^2 = \|\tilde{q}\|^2$, the unnormalized rotation matrix is

$$\tilde{R}(\tilde{q}) = \begin{bmatrix} r^2 - 2(q_y^2 + q_z^2) & (2q_x q_y - 2q_w q_z) & (2q_x q_z + 2q_w q_y) \\ (2q_x q_y + 2q_w q_z) & r^2 - 2(q_z^2 + q_x^2) & (2q_y q_z - 2q_w q_x) \\ (2q_x q_z - 2q_w q_y) & (2q_y q_z + 2q_w q_x) & r^2 - 2(q_x^2 + q_y^2) \end{bmatrix}$$

which can be normalized as $R = \tilde{R}/r^2$.

Let the update to the unnormalized quaternion be $\Delta q = (\Delta q_x, \Delta q_y, \Delta q_z, \Delta q_w)$. We consider an update in the direction of $\Delta q$ with a step size $a$, denoted as $q' = \tilde{q} + a\Delta q$ with $r'^2 = (q_x + a\Delta q_x)^2 + (q_y + a\Delta q_y)^2 + (q_z + a\Delta q_z)^2 + (q_w + a\Delta q_w)^2$

Let $R' = \tilde{R}(q')/r'^2 = R\Delta R = R(I + E)$, where $R, \Delta R, R'$ are orthogonal, and $E = R^T R' - I = R^T \tilde{R}'/r'^2 - I$. Then with the relation $\mathrm{tr}(S^{-2}\Delta R^T S^2 \Delta R) = \mathrm{tr}(S^{-1}\Delta R^T SS\Delta RS^{-1}) = \|S\Delta RS^{-1}\|_F^2 = \|S(I+E)S^{-1}\|_F^2$, we have

$$\frac{\partial}{\partial x}\|S(I+E)S^{-1}\|_F^2 = 2\mathrm{tr}((S(I+E)S^{-1})^T(S\partial_x ES^{-1})),$$

$$\frac{\partial^2}{\partial x^2}\|S(I+E)S^{-1}\|_F^2 = 2\|S\partial_x ES^{-1}\|_F^2 + 2\mathrm{tr}((S(I+E)S^{-1})^T(S\partial_x^2 ES^{-1})),$$

where the partial derivative to $x$ is short for the partial derivative to $q_x$. We have similar formulas for the partial derivative to $q_y, q_z, q_w$. For the variable $q_c \in \{q_x, q_y, q_z, q_w\}$, it can be computed that $\partial_c E = R^T(r^{-2}\partial_c \tilde{R} - 2q_c r^{-4}\tilde{R})$, and

$$\partial_c^2 E = R^T(-2q_c r^{-4}\partial_c \tilde{R} + r^{-2}\partial_c^2 \tilde{R} + 6q_c^2 r^{-6}\tilde{R}_c - 2q_c q^{-4}\partial_c \tilde{R} - 2q^{-4}\tilde{R} + 2q_c^2 r^{-6}\tilde{R})$$
$$= R^T(-2q_c r^{-4}\partial_c \tilde{R} + r^{-2}\partial_c^2 \tilde{R} + 8q_c^2 r^{-6}\tilde{R} - 2q_c r^{-4}\partial_c \tilde{R} - 2r^{-4}\tilde{R}).$$

Denote $T(\Delta q) = \|S(I+E)S^{-1}\|_F^2$, then when $\Delta q_x \ne 0, \Delta q_y = 0, \Delta q_z = 0, \Delta q_w = 0$, we have $T(\Delta q_x) = 3, T'(\Delta q_x) = 0, \beta_x = T''(\Delta q_x) = 2\|S\partial_x ES^{-1}\|_F^2 + 2\mathrm{tr}(\partial_x^2 E)$. We thus can approximate $\mathrm{tr}(S^{-2}\Delta R^T S^2 \Delta R)$ by its second-order Taylor expansion and obtain the approximation of (7) as $3 + \frac{1}{2}\beta_x(\Delta q_x)^2 - 3 < -4\ln(1 - \epsilon/\alpha)$. Rearranging the terms, we obtain

$$|\Delta q_x| < \sqrt{-8/\beta_x \ln(1 - \epsilon/\alpha)}.$$

Using similar derivation, we also obtain the bounds

$$|\Delta q_y| < \sqrt{-8/\beta_y \ln(1 - \epsilon/\alpha)}, \ |\Delta q_z| < \sqrt{-8/\beta_z \ln(1 - \epsilon/\alpha)}, \text{ and } |\Delta q_w| < \sqrt{-8/\beta_w \ln(1 - \epsilon/\alpha)}.$$

