# OpenReview forum: "3DGS$^2$-TR: A Scalable Second-Order Trust-Region Method for 3D Gaussian Splatting"
_ICML.cc/2026/Conference — ICML 2026 regular_

### Official Review · Reviewer_GvoV · 2026-03-13

**Soundness:** 3
**Presentation:** 3
**Significance:** 3
**Originality:** 2
**Overall Recommendation:** 3
**Confidence:** 3

**Summary:**

This paper outlines a method for accelerating the fitting problem for 3D gaussian splatting (3DGS) with second-order optimization. The novelty of the method is using diagonal approximation for the Hessian matrix via Hutchinson's method to keep the complexity same as Adam, while also employing a Hellinger distance-based trust region for regularization.

**Compliance With Llm Reviewing Policy:**

Affirmed.

**Final Justification:**

The authors have not addressed the questions during the discussion period, thus I keep my score of 3 (weak reject).

**Key Questions For Authors:**

Some minor points that can be addressed:
* Define $\bar\alpha$ or give a reference for the rendering equation.
* Eq 4 mentions that $f_i$ holds the square root of a loss component whereas line 216 on the right column refers to it as the residual vector. There may be some confusion on whether $f_i$ has the residual or the square root of the residual.
* Provide reference for third party implementation on line 358, 3DGS on line 361.

Some technical questions:
* How does the use of SHD compare to different regularizations on the Gaussian parameters (such as hard/soft thresholding, L1, L2 norms, total variation (TV) etc)? The authors should add an ablation study to justify the choice of SHD. What other thresholding/proximal operations/regularizations did you try? Why choose SHD? How do you support the claim that it is more interpretable or natural?
* How is the performance affected when trust region regularization is disabled completely?
* The authors suggest the apparent mass of a splat scales by moving along the viewing direction (315-317). Explain this or give references.
* Do you have a comparison of how the number of splats affect performance and optimization time for different methods? Can you provide a scatter plot of (metric vs number of splats) with optimization time as the size of the plotted points? Does the method provide the most gain for large number of splats?
* One of the claims of the paper is that the 2nd order optimization assumptions (smoothness etc) don't apply to 3DGS problem. Which parts of your designed algorithm address this specifically? Is it just the trust region clipping?
* Since 3DGS-LM has results for 35-75-150 iterations, can you share how the other methods perform at the same number of iterations?

**Limitations:**

The authors have a limitation section, which can benefit from more in depth discussion on how diagonal estimator handling insertions and translations of splats can be enabled in future work.

**Strengths And Weaknesses:**

Soundness: The submission is technically sound. The equations, derivations are well explained. Some claims can be supported by more experiments (especially the use of SHD, see questions below). Additional experiments and minor fixes to the text for improvements are outlined below.

Presentation: The explanations follow a logical flow, and it is easy to understand the proposed concept. Overall presentation is good, with potential improvements listed under questions.

Significance and originality: The GS fitting problem is relevant in the area of computer vision. This paper combines compute-efficient second-order optimization and regularization with the fitting objective, providing faster reconstruction. The choice of using diagonal estimations for the Hessian matrix is well grounded, but SHD is not motivated as rigorously. It would benefit from comparisons to other regularizers and having the baseline of not including trust region constraint at all (3DGS$^2$ without TR). The originality comes from merging existing methods. The novelty can be justified better.

---

> ### Author Rebuttal · Authors · 2026-03-30
>
> Thank you for your valuable feedback. We address all of the questions and concerns below.
>
> **How does the use of SHD compare to different regularizations on the Gaussian parameters? The authors should add an ablation study to justify the choice of SHD.**
>
> A significant advantage to our adaptive trust region method is that the geometry of the Gaussian splat is respected through only one hyperparameter. In contrast, other regularization techniques would have to be tuned per parameter type (mean, covariance, opacity, color, etc.) as they have different geometric interpretations. However, we compare SHD to a naive fixed threshold trust region radius, where we set the threshold for each parameter type to be the learning rate of the ADAM optimizer. Link: https://anonymous.4open.science/r/8D9R6H4T/exp1.md
>
> **What other thresholding/proximal operations/regularizations did you try?  Why choose SHD? How do you support the claim that it is more interpretable or natural?**
>
> The choice of SHD is not strict – any function that measures the difference between two Gaussian splats will do. We have also tried the trust region with KL divergence; the advantages of SHD are listed below.
> - Compared to total variation or L1/L2 difference, a distance function with closed form solutions for Gaussian distributions (such as KL and SHD) makes it cheap to compute directly from the splat parameters without rasterization.
> - Unlike KL divergence, SHD is symmetric between the two input splats.
> - For SHD, all differences between input splats are accumulated positively. But for KL divergence $\int P(x) \log \frac{P(x)}{Q(x)} dx$, where $P(x)$ is less than $Q(x)$, it contributes negatively to the distance.
> - Most critically, since Gaussian splats are not real probability distributions (probability mass is not 1), SHD handles the difference in mass much more naturally (section 4.2, probability mass $Z, Z'$), whereas KL only works for Gaussian splats with the same mass.
>
> **How is the performance affected when trust region regularization is disabled completely?**
>
> Thank you for raising this concern. As Hutchinson’s method has high variance for matrices with high off-diagonal entries, it can severely underestimate some of the diagonal entries at the start of training, which causes some of the parameter updates to blow up. Without the trust region clipping, Sophia by itself would very quickly jump to NaN values for many of the Gaussian splats. We will make this clear in a revision.
>
> **The authors suggest the apparent mass of a splat scales by moving along the viewing direction (315-317). Explain this or give references.**
>
> The size of the splats varies by up to orders of magnitude in the same scene. If a uniform trust region radius is applied to all the splats, then some smaller splats can be unbounded (i.e., because they are so small, no matter how they move, the SHD will not exceed the trust region).
>
> We deal with this issue by rescaling the splats in a natural way. Since a splat that is farther away from the camera will appear to be smaller than a splat that is closer to the camera, we can “normalize” a splat by viewing it from different distances based on its size.
>
> **Do you have a comparison of how the number of splats affect performance and optimization time for different methods? Can you provide a scatter plot of (metric vs number of splats) with optimization time as the size of the plotted points?**
>
> We ran an experiment where a scene is randomly initialized with a different number of splats. We show that our method always performs better than the baselines, but it is inconclusive that the number of splats affect the relative performance of the methods. Link: https://anonymous.4open.science/r/8D9R6H4T/exp3.md
>
> **One of the claims of the paper is that the 2nd order optimization assumptions (smoothness etc) don't apply to 3DGS problem. Which parts of your designed algorithm address this specifically? Is it just the trust region clipping?**
>
> Our work deals with this problem in two ways. First, we make our steps as cheap as first order steps to quickly adapt to the noisy loss landscape. Second, we make each update step small (through trust region), since only the local quadratic approximation is accurate.
>
> **Provide reference for third party implementation on line 358, 3DGS on line 361.**
>
> The third party implementation is an in-house implementation of the forward and backward automatic differentiation of 3DGS. The full github link will be provided in the final version of the paper.
>
> **Define $\bar \alpha$ or give a reference for the rendering equation.**
>
> $\bar \alpha$ is the opacity of a 2D splat for some pixel, after projecting a 3D Gaussian onto 2D. See section 4 in [1].
>
> **Eq 4 mentions that $f_i$ holds the square root of a loss component whereas line 216 on the right column refers to it as the residual vector.**
>
> These two definitions are equivalent, but we will remove one for clarity.
>
> **References**
>
> [1] Kerbl et al., 2023 (3DGS)

---

> > ### Author Rebuttal · Reviewer_GvoV · 2026-04-04
> >
> > I thank the authors for their response. Most of my concerns are resolved. In the weaknesses, I had mentioned "The originality comes from merging existing methods. The novelty can be justified better." Can the authors provide more experiments/explanations to highlight their contribution? The second order adaptation for training is incremental considering the existing methods, and the memory & compute efficiency should be documented better to justify novelty.

---

### Official Review · Reviewer_fEew · 2026-03-16

**Soundness:** 3
**Presentation:** 3
**Significance:** 2
**Originality:** 2
**Overall Recommendation:** 4
**Confidence:** 3

**Summary:**

This paper proposes a second-order optimizer for training 3D Gaussian Splatting (3DGS) models. The main technical contributions are: (1) a successful adaptation of Sophia, an efficient second-order optimizer, to the 3DGS setting, and (2) a new trust-region strategy to improve optimization robustness. Experiments on standard 3DGS datasets demonstrate improved optimization convergence and rendering quality compared with the vanilla Adam optimizer.

**Compliance With Llm Reviewing Policy:**

Affirmed.

**Final Justification:**

The authors have addressed my concerns, and I find the paper interesting for its application of improved optimizers to 3D Gaussian Splatting (3DGS). However, I also agree with other reviewers that the contribution appears somewhat incremental (e.g., adapting Sophia to 3DGS), and the implementation does not demonstrate convincing time efficiency, which raises questions about its practical utility. Overall, I lean toward a weak accept.

**Key Questions For Authors:**

* What is the core difference between Sophia and the proposed adaptation for 3DGS? A clearer explanation would help better highlight the originality of the paper.

* The description of *“parameter-linear memory scaling”* seems somewhat irrelevant or disconnected from the following discussion. I think previous optimizers such as Adam also exhibit this property, so it would be helpful to clarify what is unique in this context.

* A naive way to implement a trust-region strategy is to manually set a fixed threshold. It would be helpful to include a comparison between such a manual threshold and the proposed adaptive design to better demonstrate the effectiveness of the method.

* Is there any justification or ablation for the choice of key hyperparameters, such as $l$ and $\epsilon$?

* Likely a naive question: In line 207, why does each training image emit **6** × H × W values rather than 3 loss components? I would expect 3 to correspond to RGB.

**Limitations:**

yes

**Strengths And Weaknesses:**

* **Soundness**
  * **Strengths:** This paper adapts a second-order optimizer to 3D Gaussian Splatting (3DGS). Considering the strong nonlinearity and discontinuity in the 3DGS optimization process, developing a new optimizer is well-motivated and important. The experiments demonstrate faster optimization convergence and improved rendering quality with the proposed design, which supports the main claims of the paper.
  * **Weaknesses:** The authors propose a trust-region strategy to improve robustness and conduct an ablation by adding it to Adam, which I appreciate. However, there is no ablation that removes this component from the proposed second-order optimizer. Such an experiment would help better isolate the contribution of the trust-region design, which is important since adapting Sophia to the 3DGS setting appears to be incrementa. Furthermore, It would be helpful to include an ablation with a naive fixed threshold to compare with the proposed trust region stragety. Finally, the paper lacks justification or ablations on several key hyperparameters.

* **Presentation**
  * **Strengths:** The paper is generally well written and easy to follow. The motivation for using a second-order optimizer in 3DGS is clearly explained. The authors also provide diagrams and pseudocode to illustrate the algorithm.
  * **Weaknesses:** A minor weakness lies in the visualization. It would be beneficial to include loss curves so that readers can observe the convergence behavior more directly. Tables alone are not the most effective way to present such trends.

* **Significance**
  * **Strengths:** The paper addresses an important problem in 3DGS: accelerating and stabilizing training, which could have a meaningful impact on the 3DGS community.
  * **Weaknesses:** As mentioned in the discussion, the current implementation is still not perfect. According to Table 1, the training time remains significantly slower than Adam, which raises questions about its practical advantage. Moreover, it remains a question whether the proposed design is compatible with other acceleration techniques commonly used in 3DGS, such as initialization strategies and densification.

* **Originality**
  * **Strengths:** The paper proposes a trust-region strategy to stabilize training, which appears to be new and effective.
  * **Weaknesses:** The primary contribution: adapting Sophia to the 3DGS setting, is relatively incremental.

---

> ### Author Rebuttal · Authors · 2026-03-30
>
> Thank you for your valuable feedback. We address all of the questions and concerns below.
>
> **What is the core difference between Sophia and the proposed adaptation for 3DGS? A clearer explanation would help better highlight the originality of the paper.**
> **The primary contribution: adapting Sophia to the 3DGS setting, is relatively incremental.**
>
> We would like to clarify this point. Hessian-diagonal based optimization methods have existed for a while now [1, 2], which were developed mainly for large language model pretraining. Therefore, the main novelty of this work is not a new second-order optimization framework, but rather the application of it to 3DGS training. Additionally, we propose a trust region design that stabilizes second-order training, without which second-order training fails immediately. Whereas prior works in second-order training for 3DGS all depend on approximately solving a large linear system [3, 4], our method is the first to achieve a practical compute and memory complexity similar to first-order methods. Moreover, adapting the Hessian-diagonal estimation involves a significant amount of engineering effort in setting up the forward and backward automatic differentiation through the 3DGS rasterization, which is the main reason prior works have not been able to accomplish this.
>
> **The description of “parameter-linear memory scaling” seems somewhat irrelevant or disconnected from the following discussion. I think previous optimizers such as Adam also exhibit this property, so it would be helpful to clarify what is unique in this context.**
>
> The requirement for “parameter-linear memory scaling” is a constraint for higher-order optimization, in reference to prior works such as 3DGS-LM, which materializes the full Jacobian matrix (number of pixels by number of parameters) [3]. This restricts them from using the full set of training images in each iteration and requires them to subsample aggressively.
>
> **However, there is no ablation that removes this component from the proposed second-order optimizer.**
>
> Thank you for raising this concern. As Hutchinson’s method has high variance for matrices with high off-diagonal entries, it can severely underestimate some of the diagonal entries at the start of training, which causes some of the parameter updates to blow up. Without the trust region clipping, Sophia by itself would very quickly jump to NaN values for many of the Gaussian splats. We will make this clear in a revision.
>
> **Furthermore, It would be helpful to include an ablation with a naive fixed threshold to compare with the proposed trust region strategy.**
>
> A significant advantage to our adaptive trust region method is that the geometry of the Gaussian splat is respected through only one hyperparameter. In contrast, a fixed threshold trust region would still have to be tuned per parameter type (mean, covariance, opacity, color, etc.) as they have different geometric interpretations. To demonstrate this, we compare our method with a naive fixed threshold implementation, where the trust region radius for each parameter type is the learning rate of the ADAM optimizer. Link: https://anonymous.4open.science/r/8D9R6H4T/exp1.md
>
> **Finally, the paper lacks justification or ablations on several key hyperparameters.**
>
> We did not extensively tune the hyperparameters to achieve best performance. Most parameters used in the paper are taken from the baseline methods in order to highlight the effectiveness of our method. The EMA parameters $\beta_1 = 0.9, \beta_2 = 0.999$ come from ADAM. The rate at which the Hessian diagonals are estimated, $l = 10$, come from the SOPHIA optimizer [2]. (However, based on our experiments, $l = 15$ or $l = 20$ also works very well.) The trust region threshold uses an exponential decay scheduler following the decay rate of the original 3DGS training. The initial trust region threshold $\epsilon = 1e-6$ is the only parameter which is unique to our method. After sweeping this parameter from initial $\epsilon = 1e-5, 1e-6, 1e-7$, we found that the initial TR radius $1e-5$ yields better results than what we reported in the paper. As such, we will perform a more comprehensive sweeping and update the paper results accordingly. Link: https://anonymous.4open.science/r/8D9R6H4T/exp2.md
>
> **Likely a naive question: In line 207, why does each training image emit 6 × H × W values rather than 3 loss components? I would expect 3 to correspond to RGB.**
>
> Both the L1 loss and SSIM loss emit a 3xHxW loss image, which stack up to 6xHxW.
>
> **It would be beneficial to include loss curves so that readers can observe the convergence behavior more directly.**
>
> Thank you for this comment. We report results from the ablation studies with the loss curves and will include figures in the paper where appropriate.
>
> **Reference**
>
> [1] Yao et al., 2021 (AdaHessian)
>
> [2] Liu et al., 2023 (Sophia)
>
> [3] Höllein et al., 2025 (3DGS-LM)
>
> [4] Lan et al., 2025 (Near 2nd Order)

---

> > ### Author Rebuttal · Reviewer_fEew · 2026-04-01
> >
> > The authors have addressed all of my concerns. I appreciate the additional experimental results, which strengthen the paper and make it more convincing.

---

### Official Review · Reviewer_W7Ls · 2026-03-23

**Soundness:** 2
**Presentation:** 2
**Significance:** 2
**Originality:** 2
**Overall Recommendation:** 3
**Confidence:** 3

**Summary:**

This paper focuses on the problem of 3DGS training efficiency. They identify three principles to accelerate scene training: (1) cheap per-iteration computation; (2) parameter-linear memory scaling; (3) trust-region constraints, thus propose to use second-order optimizer for training. Specifically, they use Hutchinson's method to estimate the Hessian diagonal which is similar to a prior work, with a parameter-wise trust-region mechanism for update. This trust region is designed to constrain large or unstable updates in the highly nonlinear and piecewise-continuous 3DGS rasterization process, while respecting the geometry and semantics of individual Gaussian parameters. They evaluate on several 3DGS datasets, compared to ADAM, 3DGS-LM, and ADAM-TR.

**Compliance With Llm Reviewing Policy:**

Affirmed.

**Final Justification:**

Concern still exists as no more rebuttal reply from the authors. Would main my score.

**Key Questions For Authors:**

Please refer to the weakness part as above.

**Limitations:**

Yes.

**Strengths And Weaknesses:**

## **Strengths**
- The illustration about the motivation and method is pretty clear and detailed.
- The design of trust-region based on the representation applies closely to the application - 3DGS, I think it's very well-motivated and grounded.
- The results indicate positive signal -- consistent improvement over compared methods on SSIM, PSNR and LPIPS.

## **Weakness**
- Why are all the experiments conducted without densification? I think it is worthwhile to also show people the results of first-order methods with/without densification. Even if the results with densification might be better than your method, it is quite valuable. The results now only demonstrate effectiveness in a constrained setting.
- Limited evaluation experiments. Although the results in table.1 shows positive performance under same training epochs, the training wall-clock time is actually much longer -- 2~3X over first-order methods. And the improvement is not substantial, which makes it unclear that whether your method still wins with matched time budgets. Besides, the overall number of compared scenes is limited. Given that running on each scene is pretty fast, can the authors demonstrate on larger-scale benchmarks?
- Contribution is somewhat incremental. A part of the method is close to adapting Sophia-style optimization to 3DGS. The main novelty is therefore the trust-region design rather than an entirely new second-order framework. That is still meaningful, but causes some concerns.
- To strongly support the effectiveness of the trust-region update, can the authors design more ablation experiments to show the robustness of the derived bounds on different quantities?

---

> ### Author Rebuttal · Authors · 2026-03-30
>
> Thank you for your valuable feedback. We address all of the questions and concerns below.
>
> **Contribution is somewhat incremental. A part of the method is close to adapting Sophia-style optimization to 3DGS. The main novelty is therefore the trust-region design rather than an entirely new second-order framework.**
>
> We would like to clarify this point. Hessian-diagonal based optimization methods have existed for a while now [4, 5], which were developed mainly for large language model pretraining. Therefore, the main novelty of this work is not a new second-order optimization framework, but rather the application of it to 3DGS training. Additionally, we propose a trust region design that stabilizes second-order training, without which second-order training fails immediately. Whereas prior works in second-order training for 3DGS all depend on approximately solving a large linear system [2, 3], our method is the first to achieve a practical compute and memory complexity similar to first-order methods. Moreover, adapting the Hessian-diagonal estimation involves a significant amount of engineering effort in setting up the forward and backward automatic differentiation through the 3DGS rasterization, which is the main reason prior works have not been able to accomplish this.
>
> **Why are all the experiments conducted without densification? I think it is worthwhile to also show people the results of first-order methods with/without densification.**
>
> We demonstrate with our experiments that, given the same number of parameters and initialization, our method is able to reach a local minimum faster than first-order methods. Densification changes the loss landscape and the local minima, and since different densification methods may yield varying results, we do not wish to obfuscate the contribution of our method by including it. However, for context, we include a comparison of our method to ADAM with MCMC densification for a representative scene per dataset [6], and note that, with more parameters, the reconstruction quality of ADAM wins out. We expect the same quality once densification is integrated into our method with high confidence. Link: https://anonymous.4open.science/r/8D9R6H4T/exp4.md
>
> **Although the results in table.1 show positive performance under the same training epochs, the training wall-clock time is actually much longer -- 2~3X over first-order methods. And the improvement is not substantial, which makes it unclear whether your method still wins with matched time budgets.**
>
> The overhead of the second method (minus trust region clipping) can be exactly computed in terms of floating point operations. The diagonal estimation runs once every 10 iterations and consists of one forward and backward pass, which is an amortized 10% overhead per iteration. Our current implementation of the trust region bounds takes about 10% per iteration based on profiling results (comparing ADAM to ADAM-TR in Appendix A). Overall, with at least 50% fewer iterations, we can project a 40% improvement in training time with more efficient implementation.
>
> **Besides, the overall number of compared scenes is limited. Given that running on each scene is pretty fast, can the authors demonstrate on larger-scale benchmarks?**
>
> We evaluated our method on all the scenes and datasets in the original 3DGS paper and relevant works in the literature for accelerating 3DGS training, such as optimization-based methods [1, 2, 3], densification-based methods [6], and kernel-optimization-based methods [7]. Given that there are currently no representative datasets for large-scale 3DGS scenes, is there a particular dataset that the reviewer would like to see?
>
> **To strongly support the effectiveness of the trust-region update, can the authors design more ablation experiments to show the robustness of the derived bounds on different quantities?**
>
> An additional experiment on each derived bound is performed and the results are included here: https://anonymous.4open.science/r/8D9R6H4T/exp5.md
>
> **References**
>
> [1] Kerbl et al., 2023 (3DGS)
>
> [2] Höllein et al., 2025 (3DGS-LM)
>
> [3] Lan et al., 2025 (Near 2nd Order)
>
> [4] Yao et al., 2021 (AdaHessian)
>
> [5] Liu et al., 2023 (Sophia)
>
> [6] Kheradmand et al., 2024 (3DGS-MCMC)
>
> [7] Ye et al., 2025 (gsplat)

---

> > ### Author Rebuttal · Reviewer_W7Ls · 2026-04-02
> >
> > Thanks to the authors for the reply. The concern about the robustness of derived bound is solved. One concern still exists:
> >
> > > Therefore, the main novelty of this work is not a new second-order optimization framework, but rather the application of it to 3DGS training. Additionally, we propose a trust region design that stabilizes second-order training, without which second-order training fails immediately.
> >
> > If the main contribution is the application, I think achieving more efficient implementation of the second-order framework will be much more valuable than describing the optimization framework for 3-4 pages in the paper, given that the wall-clocked time is 2~3X than first-order method within the current implementation.
> >
> > Also, there should be some reference paper [1] evaluating 3DGS quality on large-scale datasets and diverse metrics.
> >
> > *I am not an expert in this field.
> >
> > [1] Jiang, Lihan, et al. "Anysplat: Feed-forward 3d gaussian splatting from unconstrained views." ACM Transactions on Graphics (TOG) 44.6 (2025): 1-16.

---

### Decision · Program_Chairs · 2026-04-30

**Decision:**

Accept (regular)

**Comment:**

The paper received mixed reviews.  After rebuttal, two reviewers acknowledged that their major concerns have been addressed  properly, and some minor issues can be improved, and one reviewer kept their rating unchanged.   The AC read all comments and discussion, and judges this paper has it value, and can be accepted.   In the camera ready version, please address all reviewers'c comments.